# Tunnel junctions based on interfacial two dimensional ferroelectrics

Yunze Gao [1,2,6], Astrid Weston [1,2,6], Vladimir Enaldiev [1,2,6], Xiao Li[1,2], Wendong Wang [1,2], James E. Nunn[3], Isaac Soltero [1,2], Eli G. Castanon[1,2], Amy Carl [1,2], Hugo De Latour [1,2], Alex Summerfield [1,2], Matthew Hamer[1,2], James Howarth [1,2], Nicholas Clark [1,2], Neil R. Wilson [3], Andrey V. Kretinin [1,2,4] ✉, Vladimir I. Fal'ko [1,2,5] ✉ & Roman Gorbachev [1,2,5] ✉

Van der Waals heterostructures have opened new opportunities to develop atomically thin (opto)electronic devices with a wide range of functionalities. The recent focus on manipulating the interlayer twist angle has led to the observation of out-of-plane room temperature ferroelectricity in twisted rhombohedral bilayers of transition metal dichalcogenides. Here we explore the switching behaviour of sliding ferroelectricity using scanning probe microscopy domain mapping and tunnelling transport measurements. We observe well-pronounced ambipolar switching behaviour in ferroelectric tunnelling junctions with composite ferroelectric/non-polar insulator barriers and support our experimental results with complementary theoretical modelling. Furthermore, we show that the switching behaviour is strongly influenced by the underlying domain structure, allowing the fabrication of diverse ferroelectric tunnelling junction devices with various functionalities. We show that to observe the polarisation reversal, at least one partial dislocation must be present in the device area. This behaviour is drastically different from that of conventional ferroelectric materials, and its understanding is an important milestone for the future development of optoelectronic devices based on sliding ferroelectricity.

Ferroelectrics (FE) are a class of materials with the ability to maintain a spontaneous electric polarisation that can be switched by an external electric field. Due to this, ferroelectric materials have become the key element in devices for a broad range of electronic applications including sensors, capacitors, non-volatile memory, electro-optical switching, and many others[1]. Room temperature FE devices are typically based on conventional bulk ferroelectrics, such as perovskites (e.g. $PbTiO_3$, $BaTiO_3$, $SrTiO_3$), with a minimum thickness of a few

nanometres[2–5]. While decreasing their thickness is desirable to reduce operating voltages and miniaturize their design, there are major challenges in engineering atomically thin metal-oxide ferroelectrics due to instability caused by depolarisation, interface chemistry, and high contact resistances[6].

2D materials are promising candidates for the next generation of (opto-)electronic devices with memory function due to their ultimate thickness limit and the lack of dangling bonds, making them immune

[1]Department of Physics and Astronomy, The University of Manchester, Oxford Road, Manchester M13 9PL, UK. [2]National Graphene Institute, The University of Manchester, Oxford Road, Manchester M13 9PL, UK. [3]Department of Physics, University of Warwick, Coventry CV4 7AL, UK. [4]Department of Materials, The University of Manchester, Oxford Road, Manchester M13 9PL, UK. [5]Henry Royce Institute for Advanced Materials, The University of Manchester, Oxford Road, Manchester M13 9PL, UK. [6]These authors contributed equally: Yunze Gao, Astrid Weston, Vladimir Enaldiev. ✉e-mail: andrey.kretinin@manchester.ac.uk; vladimir.falko@manchester.ac.uk; roman@manchester.ac.uk

to depolarisation[7]. In previous years, various groups have reported observations of intrinsic 2D in-plane ferroelectricity in materials such as (monolayer) SnTe[8], out-of-plane ferroelectricity in (d1T) MoTe$_2$[9], (1T') WTe$_2$[10] few-layer (Td) WTe$_2$[11] and CuInP$_2$S$_6$[12,13] and both types in In$_2$Se$_3$[14–16]. More recently, engineering the ferroelectric interfaces with broken inversion symmetry has opened a path to achieving interfacial ferroelectricity in twisted homo-bilayers of insulating hBN[17–20], exfoliated[21] and twisted semiconducting transition metal dichalcogenides (TMDs) such as bilayer MoS$_2$, MoSe$_2$, WSe$_2$, and WS$_2$[20,22,23]. In these studies, two thin crystals of 2D materials have been mechanically stacked with a small rotational misalignment (twist). This enables atomic reconstruction, resulting in large structural domains of alternating stacking order featuring broken inversion symmetry[20]. Due to asymmetric hybridisation between the conduction band states in one layer and the valence band states in the other layer, charge transfer occurs between the layers producing a built-in electric field across the van der Waals gap. Such out-of-plane ferroelectric polarisation is bound to the underlying atomic structure of each domain, with switching enabled by the sliding movement of the partial dislocations separating reconstructed domains. While multiple applications of sliding ferroelectricity have been proposed[7], an understanding of its switching behaviour in the context of electronic devices is still lacking.

Here, we study the polarisation-dependent tunneling electroresistance (TER) in ferroelectric tunnel junctions (FTJ) with a composite FE and non-polar dielectric barrier. We design the ferroelectric interface with transition metal dichalcogenides which have recently been shown to display robust sliding ferroelectricity[20,22], high crystalline and electronic quality[24], as well as outstanding optical properties[25] and are considered necessary for next-generation optoelectronic devices. We demonstrate ambipolar switching behaviour in the tunnelling current with ON/OFF ratios > 10. While methods of substantially increasing the ON/OFF ratio have been extensively demonstrated for conventional ferroelectrics[1,3–5], here we focus on the switching behaviour instead, and show that it is remarkably different. We study multiple devices positioned over various layouts of the complex domain network and reveal a strong dependence on switching behaviour due to the nature of the local domain structure.

## Results

To produce the ferroelectric interface, two monolayers of MoS$_2$ have been transferred on top of each other with marginal twist angle using the tear-and-stamp process[26] and then placed onto a thick (typically several tens of nanometers) exfoliated graphite crystal. This process results in twist angle disorder (typically ±0.1°[27–29]), a common feature of twisted bilayers assembled from isolated 2D crystals, as a consequence of the random strain produced during the transfer process. Here, the twist angle variation is more pronounced due to the use of a flexible polymer support in order to achieve a diverse domain network in each sample. An example of such a domain network visualised using lateral/friction-mode AFM with a conductive tip can be seen in Fig. 1a. The bright and dark regions correspond to domains with different local stacking order, which we denote as Mo$^t$S$^b$ (Mo atom in the top layer is aligned with S atom in the bottom layer) and S$^t$Mo$^b$ (vice versa), Fig. 1b. These structural domains possess opposite out-of-plane/perpendicular electrical polarisation with a ferroelectric potential of $\Delta V_{FE} = \pm 63$ mV and can be locally switched by lateral migration of the partial dislocations separating the domains. Furthermore, the expansion and compression of the domains can be controlled by the applied external field[20].

Four characteristic domain configurations have been selected for device fabrication as highlighted with dashed circles in Fig. 1a: (1) over a single partial dislocation between two large, roughly equal Mo$^t$S$^b$ / S$^t$Mo$^b$ domains; (2) over regular triangular domains with tens of nm period; (3) over three domains separated by domain walls; (4) similar configuration to (3) but with the middle domain fully collapsed into a

perfect dislocation. The heterostructure was then covered by a few-layer hBN flake acting as a tunnelling barrier, a top graphene (source) electrode, and a bottom graphite (drain) electrode, as shown in Fig. 1c. The graphene electrode was selectively patterned producing small tunnelling contacts (~500 nm) over the selected domain configurations 1-4, so that both tunnelling current and electric field switching behaviour can be studied locally. The resulting FTJs allow measuring a change in tunnelling current with the reversal of FE polarisation. The hBN tunnelling barrier serves to enhance FTJs performance[30] and protects the device by suppressing high tunnelling currents while allowing the application of sufficient electric fields for FE switching.

The band diagram of the resulting system, corresponding to the zero potential difference set between the source and drain electrodes ($V_{sd} = 0$) is shown in Fig. 2a. Here, the conduction band of the MoS$_2$ bilayer is $U_{MoS} \approx 0.3$ eV above the Fermi level of graphite[31], while the valence band of the hBN layer is $U_{BN} \approx -2.66$ eV below the Dirac point of graphene as determined from complementary ARPES experiments (see SI Section 2). Due to the equal potentials initially set on the source and the drain, a small carrier density emerges in graphene and graphite to compensate for the contact potential difference (0.26 eV[32]) between them, and to negate the FE potential from the rhombohedral MoS$_2$ bilayer ($\Delta_{FE} \approx 63$ meV[33,34]). This results in a corresponding electric field so that the potential profile across the dielectric stack has an additional linear contribution equalising the electrochemical potentials $\mu_1$ and $\mu_2$. Depending on the FE polarisation state of the MoS$_2$ bilayer, the combined tunnelling barrier in both MoS$_2$ and hBN can be switched to a lower or a higher state, which applies for both current directions. As the bias voltage is applied across the junction, Fig. 2b, ~60 % of the total potential $V_{sd}$ drops across the hBN (3L) and ~40 % in the MoS$_2$ bilayer according to the ratio of dielectric constants and thicknesses of these materials. This holds up to a point where the MoS$_2$ conduction band edge dips into the bias window at $V_{sb}$~0.8 V and the direct tunnelling into the bilayer becomes significant. Even though the tunnelling behaviour changes, the ferroelectric polarisation persists at finite electron densities up to ~10$^{13}$ cm$^{-2}$ [20,23]. This behaviour is indeed observed in devices where a single boundary is positioned roughly in the middle of the tunnelling contact (type 1).

In Fig. 2c we plot the differential conductance in such a device, which shows clear hysteresis between the upwards and downwards directions of the applied field. It is important to note here that no hysteresis is observed until we increase the bias voltage above ±1.2 V, corresponding to external electric fields ≈ 0.55 V/nm, above which significant domain wall movement is expected[20]. Therefore, the appearance of the pronounced switching at higher bias voltages can be attributed to the domain wall being fully expelled from the tunnelling area, creating a configuration where all the current flows through one polarised domain only. Once the V$_{sd}$ drops below -1 V, the opposite configuration is enabled, achieved by the reverse movement of the domain boundary across the tunnelling area and its expulsion on the other side, switching the FE polarisation in the entire device area. This is confirmed by a substantial change in the tunnelling current threshold on the return curve, both for positive and negative currents, $\Delta V^+_{sd} \approx 170$ mV and $\Delta V^-_{sd} \approx 140$ mV, respectively, which indicates a clear change in the tunnelling barrier height for different polarisations. The observed behaviour is consistent with the microscopic observations of the domain wall movement, where a substantial opposite field is required to completely reverse the domain layout[20]. The gradual movement of the domain wall across the device suggested by the absence of abrupt changes in the dI/dV characteristics is similar to conventional ferroelectric switching, where this phenomenon is attributed to disorder-controlled creep processes[35]. Consequently, if the voltage is repeatedly applied in one direction only (Fig. 2d), the switching is not observed as the FE polarisation remains in the same state. At higher temperatures, we observe strong activation behaviour, whereby electrons are thermally excited from the metallic graphite

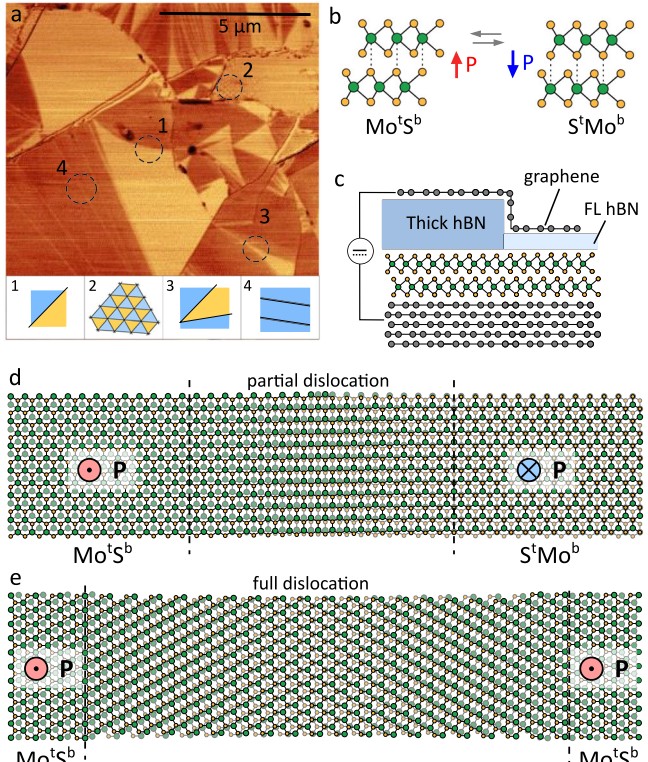

**Fig. 1 | Ferroelectric domains in rhombohedral MoS₂ bilayer. a** Contact mode friction AFM map of a marginally twisted bilayer sample on graphite with a variety of differently shaped domains present. Oppositely polarised Mo$^t$S$^b$ and S$^t$Mo$^b$ domains display clearly differentiated contrast. The black dashed circles highlight the 4 distinct domain geometries studied in this work, illustrated with matching schematics (bottom). **b** Schematics of the oppositely polarised Mo$^t$S$^b$ and S$^t$Mo$^b$ stacking configurations of the marginally twisted bilayer MoS₂ (R-MoS₂).
**c** Schematic of the ferroelectric tunnelling junction. The graphene source electrode is routed over thick (10-20 nm) hBN onto the region of interest, where the tunnelling junction with few-layer hBN and R-MoS₂ bilayer is formed. Schematic of atomic structure across a **d** partial and **e** perfect dislocation obtained using multiscale modelling [28] and TEM studies [29]. Note that this configuration is for the case when dislocations are aligned with the underlying crystallographic orientations of the TMDs, while in the sample shown in **a** these often deviate and therefore may have more complex structure.

drain into the MoS₂ since their energy separation is only $U_{MoS} \approx 0.3$ eV (see SI Fig. S5).

To understand the observed hysteresis behaviour, we model the FE tunnelling junction as a sequence of tunnel barriers with the profiles set by the band offsets and electric field distribution across the structure[36],

$$I \propto \int_0^{-eV_{sd}} e^{-2S(\varepsilon)} d\varepsilon, \; S(\varepsilon) = S_{MoS} + S_{BN},$$
$$S_X = \int_0^{d_x} \sqrt{\frac{2m_X}{\hbar^2}} \sqrt{\varepsilon_{\pm,X}(z) - \varepsilon} dz, \; X = \text{MoS, BN.}$$

(1)

Here, under the barrier action, $S(\varepsilon)$ of a current carrier with energy $\varepsilon$ is composed of contributions from hBN and MoS₂, characterized by barrier profiles $\varepsilon_{\pm,MoS}(z)$ and $\varepsilon_{\pm,BN}(z)$ (see below), with + and − signs distinguishing two polarization states (Mo$^t$S$^b$ and S$^t$Mo$^b$) of MoS₂ bilayer, and effective masses $m_{MoS}$ and $m_{BN}$, respectively. In hBN we take into account that the valence band is much closer to the graphene's Fermi level than the conduction band, in contrast to MoS₂ where the conduction band is closer to the graphite's Fermi level. We also note that the band offsets relevant for the tunneling process are

larger than the actual offsets of the valence band edge in hBN, $-U_{BN}$, and the conduction band edge in MoS₂, $U_{MoS}$, as those crystals are not aligned/commensurate with the graphitic source and drain, so that tunneling through them involves arbitrary middle areas of their respective Brillouin zones away from band edge points. Because of this, we cannot uniquely quantify the value of the offsets for both materials ($U_{BN} < 0$ and $U_{MoS}$), as well as their out-of-plane effective masses for the conduction band states in MoS₂ ($m_{MoS}$) and the valence band states in hBN ($m_{BN} < 0$). Moreover, the absolute value of tunnelling transparency would additionally depend on the matching of the conduction/valence band states in MoS₂/hBN at their interface and their contact with the graphitic electrodes.

Therefore, instead of attempting a detailed quantitative description of the *I(V)* characteristics, we focus on the qualitative trends related to the observed hysteresis behaviour due to polarization switching. That is, we expand the barriers and their respective contributions towards the tunneling action to the linear order of the bias field, $\mathscr{F}_{MoS/BN}$, in MoS₂/BN and double layer potential ($\pm \triangle$ for the two polarization states -- Mo$^t$S$^b$ and S$^t$Mo$^b$ -- of MoS₂ bilayer):

$$\varepsilon_{\pm,MoS}\left(z < \frac{d_{MoS}}{2}\right) = U_{MoS} + e\mathscr{F}_{MoS}z;$$
$$\varepsilon_{\pm,MoS}\left(z > \frac{d_{MoS}}{2}\right) = \varepsilon_{\pm,MoS}^{(c/v)}\left(z < \frac{d_{MoS}}{2}\right) \pm \triangle;$$
$$\varepsilon_{\pm,BN}(z) = U_{BN} + e\mathscr{F}_{MoS}d_{MoS} + e\mathscr{F}_{BN}(z - d_{MoS}) \pm \triangle,$$

(2)

determining the exponential dependence of *I(V)* characteristics as

$$\left. \ln \frac{dI}{dV_{sd}} \right|_{V_{sd}>0} \propto |eV_{sd}| \pm \frac{\delta_>}{2}, \; \delta_> \approx \frac{2\sigma(\theta-1)}{\theta-\sigma-2}\Delta$$
$$\left. \ln \frac{dI}{dV_{sd}} \right|_{V_{sd}<0} \propto |eV_{sd}| \pm \frac{\delta_<}{2}, \; \delta_< \approx \frac{2\sigma(\theta-1)}{\theta-\sigma+2\theta\sigma}\Delta$$

(3)

Here, + is used for Mo$^t$S$^b$ bilayer stacking and -- for S$^t$Mo$^b$, showing that Mo$^t$S$^b$ stacking promotes tunneling for both directions of the applied bias, whereas S$^t$Mo$^b$ stacking demotes tunneling, which is a result of lack of mirror asymmetry in the device architecture (Fig. 1). The size of the bias offsets, $\delta_>/e$ and $\delta_</e$, between the *I(V)* characteristics for the two FE states of MoS₂ bilayer is parametrized using the following characteristics of the two materials:

$$\sigma = \frac{d_{BN}}{d_{MoS}}\frac{\epsilon_{MoS}}{\epsilon_{BN}}, \; \theta = \frac{d_{MoS}}{d_{BN}}\sqrt{\frac{m_{MoS}U_{BN}^3}{m_{BN}U_{MoS}^3}}$$

(4)

Here, $d_{MoS} = 12.3$ Å and $d_{BN} = 3 \times 3.33$ Å $= 9.99$ Å are the thicknesses of the corresponding layers, while $\epsilon_{BN} \approx 3$ and $\epsilon_{MoS} \approx 6.2$ are their dielectric permittivities[37–39], hence, $\sigma \approx 1.68$. Without knowledge of the precise values of the band offsets and effective masses, we can assess the qualitative features of the *I(V)* characteristics such as the bias voltage asymmetry. In this regard, eq. (3) shows a systematic difference between the bias offsets (due to the FE polarization of the MoS₂ bilayer) for positive and negative voltages. In particular, in the case of $\theta \gg 1$, expected for $|U_{BN}| \gg U_{MoS}$ and $m_{MoS} \sim -m_{BN} \sim m_0$, we find that $\delta_> \approx 231$ meV $> \delta_< \approx 48$ meV which is in general agreement with the experiment (Fig. 2c). Also, we note only a 10-fold ratio between ON current values in the two polarization states of MoS₂ interface; to increase this ratio, one may try to vary hBN layer thickness, use monolayer/bilayer or bilayer/bilayer MoS₂ structures (SI Fig. S9), employ multiple ferroelectric interfaces or make bilayer devices of narrower-gap TMDs, including MoTe₂.

Having understood the effect of polarisation switching in single-domain tunnelling structures, we study devices fabricated of marginally twisted MoS₂ bilayers where local lattice reconstruction leads to a periodic triangular domain layout (type 2). Such devices show a much smaller hysteresis, $\delta/e \leq 50$ mV (regardless of the bias voltage sweep range), Fig. 3a. This indicates that while the domain network does

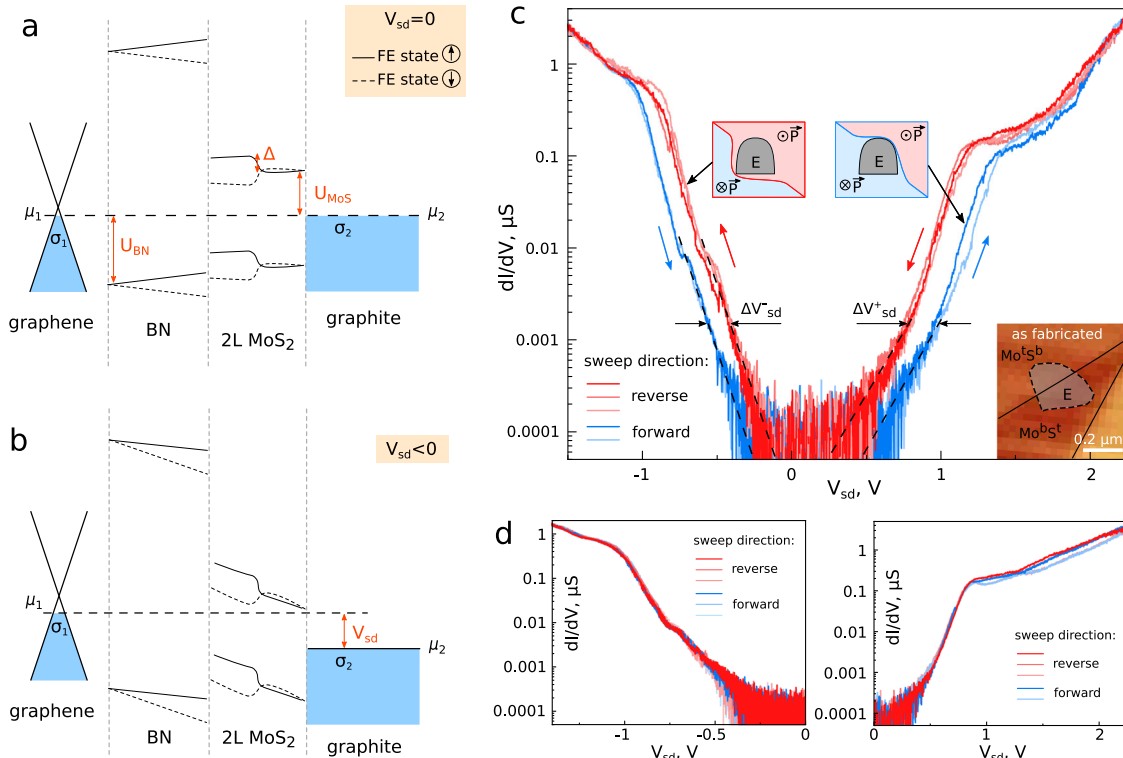

**Fig. 2 | Switching behaviour of FTJ over a Mo$^t$S$^b$/S$^t$Mo$^b$ dislocation (configuration 1) enabling full domain switching within the tunnelling area. a** Schematic band diagram with equipotential source and drain, $V_{sd}$ = 0. A small charge density σ is induced in graphene and graphite to cancel out the contact potential difference and ferroelectric potential from the rhombohedral MoS$_2$ bilayer. Bold and dashed lines illustrate the potential profile created for upward and downward polarisation respectively. **b** Schematic band diagram when applying a small reverse bias $V_{sd}$, before the conduction band states of MoS$_2$ become available for tunnelling. **c** Tunnelling conductance (dI/dV) as a function of the transverse electric field ($V_{sd}$) between the graphene source and the graphite drain. The direction of the $V_{sd}$ sweeps is indicated by the arrows. Schematic insets show the domain configuration that produces the observed tunnelling behaviour, where the area of the FTJ electrode is indicated in grey (E). In the bottom-right corner, a friction AFM map shows the existing domain configuration prior to the sweeps. For this device, the tunnelling hBN has a thickness of 3 layers. These results were acquired at T = 1.5 K. **d** Same as **c** but with the field only applied in one direction showing that no polarisation reversal is occurring within the sample. No hysteresis is observed in these cases.

experience some changes due to expansion/contraction of Mo$^t$S$^b$/S$^t$Mo$^b$ areas, complete switching cannot be achieved, since both types of polarisation are present in the tunnelling area at all times. This complies with the earlier observed transformations of domains/domain wall networks in marginally twisted MoS$_2$ bilayers, where it was noted that the networks of domain walls separating areas with opposite FE polarisation have a higher rigidity in bilayers with shorter moiré periods, and nodes of such networks are essentially pinned[40,41] due to $C_3$ symmetry of the acting forces. Also, we note that the period of the network of partial dislocations – domain walls is set by the local twist angle, imprinted in the device by the transfer, encapsulation, and contact deposition.

Similar behaviour is observed in a sample where two domain walls are present and connected to an adjacent intersection, Fig. 3b. In this sample, however, the tunnelling junction was defined not only in graphene, but also etched into the MoS$_2$ bilayer, removing the ferroelectric material around the junction. While we can only see the initial domain configuration, we expect the domain configuration to be pinned at the boundary of the tunnelling area due to the etching process, known to introduce substantial damage in the vicinity of the edge[42]. Therefore, the partial dislocation cannot be fully expelled from the sample. Similar pinning behaviour was recently observed on other structural defects, such as cracks, edges and contamination[20,43].

The application of ferroelectric materials in electronic devices depends on one's ability to switch the polarisation states of ferroelectric layer. The repolarisation can be considered as a two-stage process starting from nucleation of seeds, separated by domain walls

from oppositely-polarised surrounding, followed their expansion into mesoscopic domains. Note that these domain walls are nothing but partial dislocations in rhombohedral layered crystals and are examples of topologically stable defects. The above-discussed examples of FE switching/hysteresis behaviour all include tunnel junctions for which domain nucleation did not happen as the areas under the source contained both Mo$^t$S$^b$ and S$^t$Mo$^b$ domains, so that the repolarisation of the structure required only motion or elastic deformation of the pre-existing domain walls. Therefore, to test an opportunity of the seed nucleation, we also investigated tunnelling through single polarisation domain areas of MoS$_2$ bilayer, both pristine and pierced by one perfect (full) dislocation, joining domains with the same polarisation (Fig. 3c). In the fabrication of such devices, we used a 7 L hBN tunnelling barrier enabling application of a higher electric bias for a wider range control of the ferroelectric state. However, despite applying stronger out-of-plane electric fields up to 0.85 V/nm, we systematically do not observe any hysteresis in the measurements. This behaviour qualitatively differs from the observations made on conventional FE materials, such as Pb(Zr,Ti)O$_3$ and BaTiO$_3$ where nucleation of domains with inverted FE polarisation was observed[44,45] in electric fields as low as ∼0.01 V/nm.

We attribute this difference in behaviour to the sliding nature of the polarisation switching, specific to van der Waals systems with interfacial ferroelectricity[17–20], which involves high elastic energy costs on domain wall bending to encompass a nanoscale seed with opposite polarisations. Below we consider several scenarios for a seed nucleation, including those tested in experiment, and evaluate for each of them a critical linear size of a seed that satisfies energetical trade-off

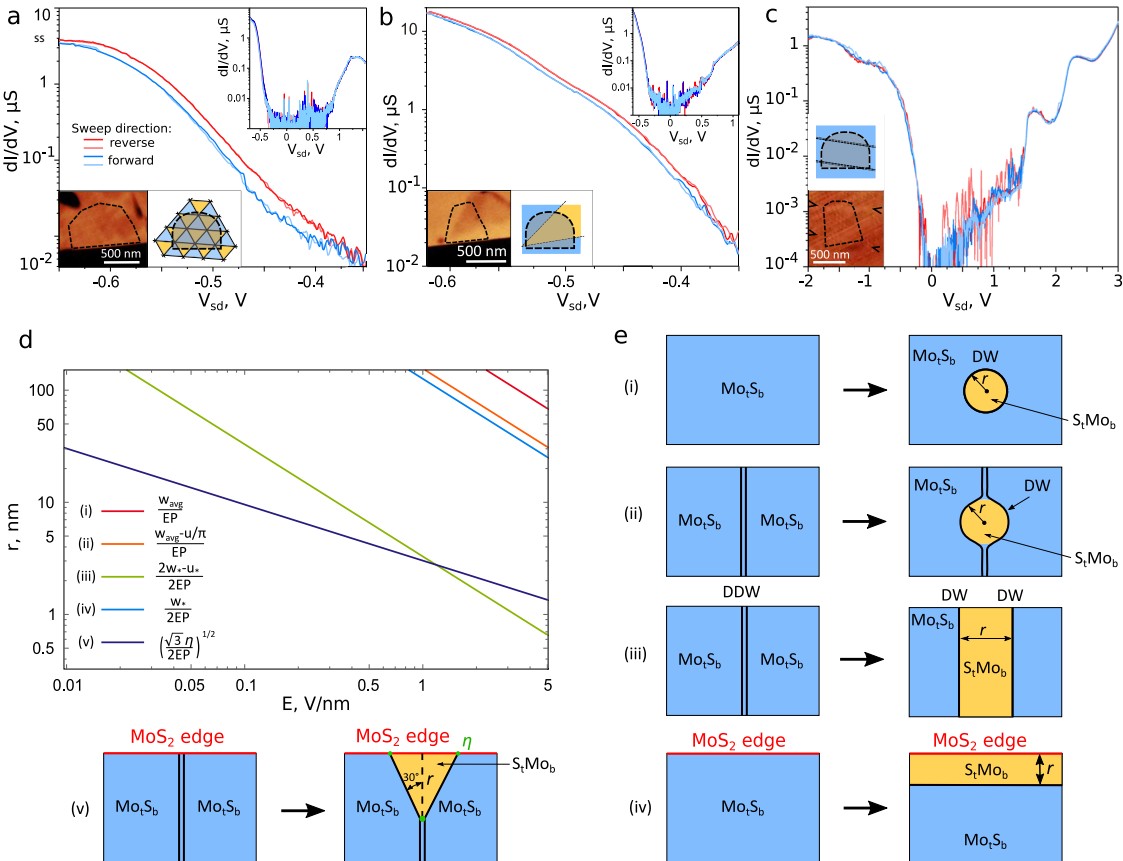

**Fig. 3 | Polarisation switching in FTJ with different domain layout and various nucleation scenarios.** Tunnelling conductance (dI/dV) as a function of the transverse electric field (Vsd) between the graphene source and the graphite drain for: **a** a periodic triangular domain network (L-60–150 nm) **b** three domains with domain walls pinned at the edges of tunnelling area using RIE and (**c**) uniform domain area with two perfect dislocations crossing the studied region. Tunnelling hBN thickness is 2 layers (**a**, **b**), and 4 layers (**c**). This hysteresis is only observable for **a** and **b**, reaching 17 mV in **a** and 7 mV in **b** between the forward and reverse sweeps. For the devices in **a** and **b**, the underlying hBN and R-MoS$_2$ were etched as well as the top graphene contact to eliminate the possible interaction between devices and the surrounding unbiased bilayer. All the results were acquired at $T = 1.5$ K. Insets show the pre-existing domain configuration visualised using friction AFM (left) and its schematic representation (right) with colour indicating domain polarisation (blue and yellow for in and out of plane) and solid lines showing dislocations. **d** Dependences of critical size of a reversal polarisation domain seed on out-of-plane electric field for five scenarios of nucleation shown in **e**. In case (v) we estimated $\eta = 0.04$ eV as configuration-averaged energy (-0.4 eV/nm [2,28]) of stacking fault with the area of MoS$_2$ monolayer unit cell.

between costs of extending length of domain wall and its bending as compared to the gain due to polarisation reversal inside the seed.

Below, we compare various options for how repolarisation can develop in the studied devices. First, we consider an isolated circular seed of S$^t$Mo$^b$ stacking inside a Mo$^t$S$^b$ domain (case (i) in Fig. 3e). Such seed will have a critical radius when it becomes energetically favourable, $r = w_{avg}/EP$, determined by $w_{avg} = 1.305$ eV/nm, the orientation-averaged domain wall energy per unit length[34] and $E$, the repolarising field. Such critical nucleation radius would be smaller, $r = (w_{avg} - u_*/\pi)/EP$, for a seed initiated at a perfect dislocation (case (ii) in Fig. 3e), characterised[34] by an energy per unit length of $u_* = 2.24$ eV/nm. Here, we use the experimentally-confirmed[32,38] theoretical estimations[18] that domain wall energy per unit length is ~1 eV/nm, determined by the energy cost of strain. We also note that such large domain wall energy protects the studied devices against dipolar instability, as for the interfacial FE polarisation studied here, the dipole-dipole interaction for domains with a size L is equivalent to only $\sim 6\frac{\mu eV}{nm}\ln\left(\frac{L}{nm}\right)$ reduction of domain wall energy (see in SI Section 6). Alternatively, one can imagine repolarisation to happen via unzipping a perfect dislocation in a pair of parallel partial dislocations (case (iii) in Fig. 3e): which would have energetically unfavourable orientations with energy per unit length $w_* = 1.13$ eV/nm. This determines the critical width for the repolarised stripe as $r = (2w_* - u_*)/2EP$ (stability of an optimally-oriented perfect dislocation at $E = 0$ against such

splitting has been shown in Ref. 18, whereas individual transfer-induced perfect dislocations would be pinned in a narrower part of the flake by simply their shorter geometrical length). The above three scenarios of a critical seed formation are illustrated in Fig. 3d to show that forming a seed with size comparable to twice a typical domain wall width would require a high field, $E \sim 1$ eV/nm. Together the cases (i), (ii) and (iii) provide relatively high energy cost of the spontaneous domain nucleation, which explains lack of switching in devices where no, or only a perfect dislocation exist, such as in Fig. 3c. We would like to point out that this is not the case when two partial dislocations are pushed together by an external field and have been subsequently shown to split upon the field reversal[20,23]. This is likely due to incomplete merger of the partial dislocations seen in[20] which provides seed for the reverse switching, as opposed to our case where the perfect dislocation is fully formed during the sample fabrication.

Potentially, seeding the inverted polarisation may be promoted at the physical edge of the MoS$_2$ bilayer. One option is that a repolarised domain forms as a stripe (case (iv) in Fig. 3e) with the energy cost determined by a single domain wall energy. This energy cost would be minimised for a domain wall oriented following the armchair direction in the crystal, $w = 0.96$ eV/nm, with the corresponding critical stripe width being $r = w/2EP$. This scenario is slightly less restrictive than the polarisation reversal inside a homogeneous domain. In a flake pierced by a full dislocation the nucleation of reversed polarisation appears to

be even easier if it happens at the point where the full dislocation reaches the crystal's edge. As an estimate we consider an armchair edge with a perfect screw dislocation sticking along zigzag direction (case (v) in Fig. 3e). Then, the repolarisation seed unzips this dislocation, forming a triangular seed with sides oriented along the other two armchair axes (lowest energy direction for $Mo^tS^b/S^tMo^b$ domain wall). The energy of the triangular seed with a width $r$ is $-\frac{2}{\sqrt{3}}EPr^2 + 2w\frac{2r}{\sqrt{3}} - u_*r + \eta$, where $\eta$ characterises energies of stacking faults at the edge termination and merging site of individual dislocations. We note[34] that, accidentally, $\frac{4w}{\sqrt{3}} \approx u_*$, so that we arrive at conditions for the critical seed size, $r = \sqrt{\sqrt{3}\eta/2EP}$, which are relaxed when compared to all the other cases, Fig. 3d, over a broad range of energy $\eta$. Therefore, we conclude that crystal edges would play a major role in polarisation switching. Further investigations of the domain wall textures near the edges are required, as well as their dependence on crystallographic orientation, and functionalization or passivation of dangling bonds. To realise such studies, one would need to design a suitable device architecture that would avoiding direct tunnelling currents near the R-TMD bilayer edge.

In conclusion, we observe drastically different switching behaviour to that of classical ferroelectric materials, which is extremely sensitive to the location of the FTJ. To achieve full switching, a single domain boundary must pre-exist within the device area and the ferroelectric material must extend beyond the source electrode footprint, providing room for the domain boundary to exit from the junction. Other domain layouts enable partial switching, which nevertheless has merit and can be considered for applications where continuous resistive switching can be employed. Finally, the devices with no domain boundaries or those only containing perfect dislocations do not display switching behaviour, however our modelling indicates that a presence of the physical edge of the ferroelectric crystal within the tunnelling can provide a nucleation point and make the switching possible.

## Methods

For SPM domain mapping, Electrical AFM techniques such as piezoresponsive force microscopy (PFM) and electrostatic force microscopy (EFM) were used. Here, equally high-domain-contrast was achieved using lateral/friction and tapping mode when conductive (budget sensors Multi-75G) probes were used.

All electrical transport measurements were performed at 1.5 K (see SI for higher temperature data). When measuring the differential tunnelling conductance as a function of the transverse field, a sweeping DC bias with a constant small AC bias (1 mV) were applied to the graphene source electrode and the drain current was measured by a SR860 lock-in amplifier using a SR560 current preamplifier. The tunnelling conductance is defined as the ratio between the AC component of the tunnelling current and the applied AC bias.

## Data availability

Raw images and underlying data are available to download https://doi.org/10.48420/25586589.v1. No custom code has been used in this work.

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

## Acknowledgements

We acknowledge support from European Union's Horizon 2020 Research and Innovation programme: European Graphene Flagship EC-FET Core3 Project (grant agreement no. 881603 R.G., A.K. and V.F.), European Quantum Flagship Project 2DSIPC (820378, R.G. and V.F.), ERC Consolidator QTWIST (no. 101001515, R.G.), and the Royal Society (R.G). In addition, we acknowledge support from EPSRC (grant numbers EP/V007033/1 R.G. and V.F., EP/V036343/1 R.G. and A.K., EP/S030719/1 R.G. and V.F., EP/T027207/1 N.R.W.), the Lloyd Register Foundation Nanotechnology Grant, CDT Graphene-NOWNANO A.C., H.dL., Y.G., and China Scholarship Council (CSC) under grant numbers 201806280036 W.W, and 201908890023 Y.G. This project was supported by the Henry Royce Institute for Advanced Materials, funded through EPSRC grants EP/R00661X/1, EP/S019367/1, EP/P025021/1 and EP/P025498/1.

## Author contributions

R.G. and A.K. conceived the study. A.W., X.L. and W.W. produced experimental samples with the help from A.C., H.dL., M.H. and J.H.; Y.G. and X.L. performed electrical measurements with help from A.K.; A.W. and E.G.C. developed and performed SPM domain mapping. J.E.N. and N.R.W. performed and analysed ARPES measurements; V.E., I.S. and V.F. conducted theoretical modelling; R.G., V.F., Y.G., A.W., A.S., N.C. and A.K. wrote the manuscript. All authors contributed to the discussions and commented on the manuscript. Y.G., A.W., and V.E. have contributed to this work equally and can claim their leading role in job applications, interviews, and talks.

## Competing interests

The authors declare no competing interests.
