## [Peer Review File · Nature Communications]

Tunnel junctions based on interfacial two dimensional ferroelectricsREVIEWER COMMENTS

Reviewer #1 (Remarks to the Author):

The authors explore the switching behavior of sliding ferroelectricity in MoS₂ using scanning probe microscopy domain mapping and tunnelling transport measurements and found that the switching behavior is strongly influenced by the underlying domain structure. Here are some comments:

In Line 45, In previous years, various groups have reported observations of intrinsic 2D in-plane ferroelectricity in materials such as (monolayer) SnTe, and out-of-plane ferroelectricity in (d1T) MoTe₂, (1T') WTe₂, few-layer (Td) WTe₂ and CuInP₂S₆. 2D Materials like In₂Se₃ with out-of-plane and in-plane polarization phase needs to be cited here.

Can authors show the atomic structures of dislocations between up and down domains in experiments? Do they have the same structure as dislocation in region 4? Because when modelling a twist system, the up and down domains are separated by special stacking region rather than a traditional dislocation.

Can the movement of dislocations after applying voltage be characterized by AFM in region 1 and 2?

In region 2, the authors claimed that with shorter moire period the opposite polarization will be more rigid. However, in Figure 3a, the bias voltage is smaller than that in region 1. Can authors explain the reasons?

In Figure 3a and Figure 3b, can positive bias voltage show hysteresis?

The authors theoretically analysis different cases to grow a reversal polarization domain with out-of-plane electric field. Can any of the cases be verified in your sample?

Reviewer #2 (Remarks to the Author):

In the manuscript of Y. Gao et al., a tunneling junction based on R-stacking MoS₂ was constructed and measured at 1.5 K. The hysteretic conductance curves are analyzed through brief modelling for various ferroelectric domain structures.

Sliding ferroelectricity is a recent breakthrough in the community of ferroelectrics. A tunneling device based on the novel ferroelectric is conceptually novel. However, from the presented results one cannot see the promise of such devices (even in the perfect case, configure 1 in Fig. 2). Details are as follows.

1. The conductance ratio between two polarized states is approximately 10. It is expected to be even smaller at room temperature. The temperature dependence of device performance should be presented, in particular, at room temperature.
2. A basic question is, why it is so small. Compared with similar tunneling junctions based on In₂Se₃ or CIPS, such a result seems less appealing. For materials beyond R-stacking bilayer MoS₂, is there any hope to improve the result? Practical measures to enhance the conductance ratio should be discussed, since one cannot see immediate results from ref. 1, 3-6 as cited in page 2, line 2.

3. Page 4, paragraph 2: the contact potential difference is claimed to be 0.26 eV. This value corresponding to the difference between graphene and bulk graphite. For the monolayer and few-layer graphene used in the manuscript, the difference in work function should be much smaller.
4. Page 6: the domain wall movement is claimed to be gradual. From literatures, it is known to be quite abrupt, especially for the low frequency sweeping of electric field used both in literatures and this work. Is it possible that the abrupt change is smeared out due to the rough signal?
5. In the model of Fig. 2, the quantum capacitance of monolayer graphene seems to be omitted. Can the author estimate the Fermi level variation during voltage sweeping and justify the omission?
6. Only one sample for each type is investigated. Whether the quantitative result is robust should be clarified.

BTW, there are some mistakes to be corrected.

1. Page 2, the end line: unipolar or ambipolar?
2. Page 6, line 2: 0.170 and 0.140 mV? Or 170 and 140 mV?

Reviewer #3 (Remarks to the Author):

The manuscript "Tunnel junctions based on interfacial 2D ferroelectrics" by Y. Gao, A. Weston, V. Enaldiev et al. addresses one of the key practical challenges concerning 2D-interfacial ferroelectrics. These findings would have an immediate impact on the research efforts in the community and will ease the path for development, particularly along the application front. The results are interesting and considerable for publication in Nat. Commun. after a few semi-minor clarifications.

Comments:

The novelty of the current manuscript lies in (1) the efforts of studying the individual ferroelectric domain-based FTJs in the reconstructed superlattice rather than a macroscopic average, which has been the case so far. On the other hand, domain wall motion in such structures has been experimentally demonstrated earlier, even by the same group. A few examples are: Fig.3 Vizner et al. Science; Fig.1-2 Weston et al. Nat. Nanotechnol.; SI Fig.S2 Deb et al. Nature (all are cited in the manuscript). Therefore, regarding ferroelectric switching by sliding, the most appealing aspect of the current manuscript is: (2) the **understanding** (than the observation itself) it imparts on - what factors influence the interlayer sliding in the ferroelectric bilayers. This is certainly missing the existing literature. But, the description of the relevant phenomena is rather ill-described in the main text (detail in my suggestions). Please note that by no means I am undermining the observation. Will the work be of significance to the field and related fields?

-Yes

Does the work support the conclusions and claims?

-Yes

Are there any flaws in the data analysis, interpretation, and conclusions?

-The authors have provided adequate qualitative justification for their observations

Is the methodology sound? Does the work meet the expected standards in your field?

-Yes

Specific suggestions:

1. Page4. Use of 'shortcut'. Is it a typo? If not, please explain.
2. The DC equivalent of the circuit, used for tunneling current measurement (shown in Fig.1c), is analogous to gating the ferroelectric MoS₂ with respect to top graphene. How would the electrostatic doping affect the inbuilt dipole moment in R-MoS₂? Does it depolarize the ferroelectric crystal?
3. The theory describing seeding size vs. favorable energy, based on competitive domain-wall stiffness, deformation energy, and electrostatic energy, lacks an introduction. No physical description has been given besides citing Ref.29. I believe an intuitive description and reasoning are very much needed to make the current manuscript self-contained and would help increase the comprehensibility among a broader class of readers.
4. Domains of type 3-4 (in ref. to Fig.1) do not show any prominent hysteresis in di/dv (Fig. 3b-c). However, it is contrasting (at least partially) to earlier results. See, for example - Fig.1 Weston et al. Nat. Nanotechnol. 2022 -analogous to type 4; SI-Fig.S2 Deb et al. Nature 2022 -analogous to type 3.
The question occurs: Do the di/dv plots indicate (i) NO sliding & NO hysteresis or (ii) only NO hysteresis or (iii) something else?
5. Please consider adding a summarized conclusion at the end.
6. Usually, such device fabrication involves various thermal annealing steps, both after tear-and-stack (before lithography) and after lithography. If this is the case, comments in supplementary would be helpful for the work to be followed and reproduced in the future.
7. Please have a look at Liang et al. Shear Strain-Induced Two-Dimensional Slip Avalanches in Rhombohedral MoS₂, Nano Lett. 2023, 23, 15, 7228. Consider citing it if you find it relevant and useful.

REVIEWER COMMENTS

To all: We would like to thank the Referees once again for their helpful comments and constructive criticism during this process. We appreciate the time it has taken to review our manuscript and believe that the manuscript has been considerably improved as the result of their work. A point-by-point response to the reviewers' concerns follows.

Reviewer #1 (Remarks to the Author):

The authors explore the switching behavior of sliding ferroelectricity in MoS₂ using scanning probe microscopy domain mapping and tunnelling transport measurements and found that the switching behavior is strongly influenced by the underlying domain structure. Here are some comments:

In Line 45, "In previous years, various groups have reported observations of intrinsic 2D in-plane ferroelectricity in materials such as (monolayer) SnTe, and out-of-plane ferroelectricity in (d1T) MoTe₂, (1T') WTe₂, few-layer (Td) WTe₂ and CuInP₂S₆". 2D Materials like In₂Se₃ with out-of-plane and in-plane polarization phase needs to be cited here.

We apologise for this omission, the references [14,15] have now been added to the manuscript.

Can authors show the atomic structures of dislocations between up and down domains in experiments? Do they have the same structure as dislocation in region 4? Because when modelling a twist system, the up and down domains are separated by special stacking region rather than a traditional dislocation.

The referee is correct, the dislocations in sliding ferroelectricity are substantially different from those in traditional ferroelectrics, and of course Cases 1 and 4 also differ. To make it clear to the readers we have now included atomic schematics for both dislocation types in Figure 1d,e. These are obtained using multiscale modelling [27] and confirmed using earlier TEM studies [28].

Can the movement of dislocations after applying voltage be characterized by AFM in region 1 and 2?

Due to the (semi)metallic nature of the top graphene electrode, it is not possible to see the TMDs domain structure underneath. For this reason, we have performed the SPM measurements before the graphene was placed and designed the devices keeping this pre-existing domain layout in mind.

In region 2, the authors claimed that with shorter moiré period the opposite polarization will be more rigid. However, in Figure 3a, the bias voltage is smaller than that in region 1. Can authors explain the reasons?

The full range of the bias voltage as well as the produced electric field are determined by the overall thickness of the dielectric stack, which is different for each device studied. Also, due to the exponential nature of the electrical currents we tend to limit the V_{sd} to the value where full current

reaches 1 μ A to prevent the effects of local heating and damage to the devices. For the device in Fig.1 the hBN thickness was 3L and for the device in Figure 3a,b only 2L which is the reason Vsd ranges differ.

In Figure 3a and Figure 3b, can positive bias voltage show hysteresis?

We apologise for this omission - due to the limited space we only showed the negative bias side in the main text. The full data sets have now been added to the Supplementary Figure S4. Due to the transport gap being larger on the positive side, a complete switching is achieved by the time a measurable current can be detected at $V_{sd} \sim 0.9$ V, and further increase in the V_{sd} creates no changes in the domain structure where measurable current is present. This is a consequence of a specific hBN dielectric thickness used and the fact that domains are switched by the electric field, but the switching can only be probed when a measurable tunnelling current is present.

The authors theoretically analysis different cases to grow a reversal polarization domain with out-of-plane electric field. Can any of the cases be verified in your sample?

We apologise for being unclear; the cases (i), (ii) and (iii) in Fig. 3e are directly related to the interpretation of the experimental results shown in Fig. 3c - we use these to explain the lack of pronounced switching behaviour. In cases (iv) and (v), we propose other structures that can enable the switching in devices with no or only perfect dislocations. However, these structures are not feasible within the approach we use due to the presence of the TMD flake edge in the tunnelling region, which will cause high tunnelling currents in the region where no TMDs are present. We have added a sentence to the main text to clarify this.

Reviewer #2 (Remarks to the Author):

In the manuscript of Y. Gao et al., a tunneling junction based on R-stacking MoS₂ was constructed and measured at 1.5 K. The hysteretic conductance curves are analyzed through brief modelling for various ferroelectric domain structures.

Sliding ferroelectricity is a recent breakthrough in the community of ferroelectrics. A tunneling device based on the novel ferroelectric is conceptually novel. However, from the presented results one cannot see the promise of such devices (even in the perfect case, configure 1 in Fig. 2). Details are as follows.

1. The conductance ratio between two polarized states is approximately 10. It is expected to be even smaller at room temperature. The temperature dependence of device performance should be presented, in particular, at room temperature.

As the referee suggested we have conducted the measurements at high temperatures (100K and 150K) and can confirm that the switching is present but indeed with a smaller hysteresis. This is a consequence of the close proximity of MoS₂ conduction band edge to the Fermi level of graphite (~0.3 eV), which at elevated temperatures leads to the thermal excitation and doping of the MoS₂ conduction band. We believe that the presence of this finite electron density in MoS₂ screens the ferroelectric potential and most likely is the main reason why the hysteresis is smaller. This thermal activation also effectively reduces the tunnelling distance (the tunnelling now takes place from MoS₂ to graphene across hBN barrier only), which is evident in the dramatic increase of the tunnelling currents observed (over two orders of magnitude at 100K) unlike for the temperature-assisted tunnelling through hBN (e.g. <https://www.nature.com/articles/srep21168>).

Our choice of MoS₂ for this study was due to the preceding works [22], where the ferroelectric potential for 3R MoS₂ was first experimentally measured, which made it the best-understood system to explore FTJ devices. This close alignment is a unique feature of MoS₂, as the other TMDs have completely different band alignments with graphite (e.g. DOI: 10.1126/sciadv.1601832 and doi.org/10.1038/s41586-019-1402-1). Because of this, other TMDs should have different temperature-dependent behaviour. However, we would prefer to leave other materials out of this manuscript for two main reasons: (1) the key conclusions of this work - the changes in the switching behaviour based on the underlying domain structure – will hold, as these are based on the properties of the domain walls and (2) these devices are notoriously hard to fabricate which would require up to a year until the results on other systems are available.

To address the referee's concern, the temperature dependence, along with the relevant discussions, have been added to the SI Figure S5 and the main text.

2. A basic question is, why it is so small. Compared with similar tunneling junctions based on In₂Se₃ or CIPS, such a result seems less appealing. For materials beyond R-stacking bilayer MoS₂, is there any hope to improve the result? Practical measures to enhance the conductance ratio should be discussed, since one cannot see immediate results from ref. 1, 3-6 as cited in page 2, line 2.

We agree with the referee that the ON/OFF ratio is not as high as that demonstrated for the latest conventional ferroelectric FTJ devices. We do not optimise the ON/OFF value - this is the first study of sliding FTJ and at this point the system is not understood well enough to deal with the memory

window optimisation. Our model (Eq (2) in the main text) predicts the value of the hysteresis to be dependent on several parameters, such as the relative band alignments of the TMDs, hBN and the contacts, the material thicknesses and the ferroelectric potential Δ . We are confident the ON/OFF ratio can be improved in future follow-up studies in a similar manner to what was done for conventional ferroelectric FTJ over the years. One simple way would be to multiply the number of ferroelectric interfaces (as traditional thin film ferroelectrics are typically few nm thick and, therefore, accumulate a much larger potential difference). For instance, Nature 612, 465 (2022) shows that six 3R TMD interfaces show cumulative polarization of 330mV versus 63 meV for the single interface we study in this work.

Instead, in this study, we focus on the switching behaviour itself, which, being drastically different in the sliding ferroelectrics, certainly needs investigating first to understand the behaviour of this novel system. We also developed a simple model which qualitatively explains the observed behaviour and allows other groups to make educated guesses as to how the ON/OFF ratio can be optimised in the future. We, however, would like to refrain from speculations regarding specific pathways to optimise the ON/OFF ratio as these would not be supported by experimental evidence which other referees may request then.

3. Page 4, paragraph 2: the contact potential difference is claimed to be 0.26 eV. This value corresponding to the difference between graphene and bulk graphite. For the monolayer and few-layer graphene used in the manuscript, the difference in work function should be much smaller.

Published literature has different observations as to how quickly the work function of graphite saturates with the number of layers, but the majority report that at >10L the bulk values are seen e.g. [10.1016/j.diamond.2019.107576](https://doi.org/10.1016/j.diamond.2019.107576) or [10.1088/0953-8984/29/3/035003](https://doi.org/10.1088/0953-8984/29/3/035003). In our work, the bottom graphite drain is not a few-layer one, but has been selected to be thick (~50 nm or ~150L) and therefore we consider it to be bulk. The clarification has been added to the main text.

4. Page 6: the domain wall movement is claimed to be gradual. From literatures, it is known to be quite abrupt, especially for the low frequency sweeping of electric field used both in literatures and this work. Is it possible that the abrupt change is smeared out due to the rough signal?

Current studies, e.g. an excellent work on in-situ switching <https://www.nature.com/articles/s41563-023-01595-0#Sec13> indeed report abrupt switching for large domains and a gradual process for the smaller ones. We do occasionally see abrupt jumps in the tunnelling curves (see, for instance, SI Fig S5a) but these vary between repetitive sweeps. Most likely this is due to the random nature of the disorder pinning the domain wall propagation, which in our case leads to mixed behaviour (gradual with some random jumps).

5. In the model of Fig. 2, the quantum capacitance of monolayer graphene seems to be omitted. Can the author estimate the Fermi level variation during voltage sweeping and justify the omission?

We apologize for this omission, in the new version of SI we took into account the quantum capacitance of graphene in equations for electric field in MoS2 and hBN, which also slightly modifies

the equation for tunnelling current. This, however, turned out to be a small correction which is illustrated in the new Supplementary Fig. S7 where both are presented.

6. Only one sample for each type is investigated. Whether the quantitative result is robust should be clarified.

We would like to point out that all 5 devices studied fit into the overall understanding of the system developed here, even though some specific configurations have not been replicated. We have now added one more device to the SI which shows consistent behaviour with the sample in Fig. 3b of main text – we hope this satisfies the referee's comment.

BTW, there are some mistakes to be corrected.

1. Page 2, the end line: unipolar or ambipolar?
2. Page 6, line 2: 0.170 and 0.140 mV? Or 170 and 140 mV?

We apologize for these omissions; they have been corrected in the new version.

Reviewer #3 (Remarks to the Author):

The manuscript "Tunnel junctions based on interfacial 2D ferroelectrics" by Y. Gao, A. Weston, V. Ewaldiev et al. addresses one of the key practical challenges concerning 2D-interfacial ferroelectrics. These findings would have an immediate impact on the research efforts in the community and will ease the path for development, particularly along the application front. The results are interesting and considerable for publication in Nat. Commun. after a few semi-minor clarifications.

Comments:

The novelty of the current manuscript lies in (1) the efforts of studying the individual ferroelectric domain-based FTJs in the reconstructed superlattice rather than a macroscopic average, which has been the case so far. On the other hand, domain wall motion in such structures has been experimentally demonstrated earlier, even by the same group. A few examples are: Fig.3 Vizner et al. Science; Fig.1-2 Weston et al. Nat. Nanotechnol.; SI Fig.S2 Deb et al. Nature (all are cited in the manuscript). Therefore, regarding ferroelectric switching by sliding, the most appealing aspect of the current manuscript is: (2) the **understanding** (than the observation itself) it imparts on - what factors influence the interlayer sliding in the ferroelectric bilayers. This is certainly missing the existing literature. But, the description of the relevant phenomena is rather ill-described in the main text (detail in my suggestions). Please note that by no means I am undermining the observation. Will the work be of significance to the field and related fields?

-Yes

Does the work support the conclusions and claims?

-Yes

Are there any flaws in the data analysis, interpretation, and conclusions?

-The authors have provided adequate qualitative justification for their observations

Is the methodology sound? Does the work meet the expected standards in your field?

-Yes

We are grateful to the Reviewer for the complements and overall positive evaluation of our work.

Specific suggestions:

1. Page4. Use of 'shortcut'. Is it a typo? If not, please explain.

We apologize for the confusion, in this case we mean that connecting the measurement circuit and setting $V_{sd} = 0$ is equivalent to making an electrical short-circuit, i.e. connecting source and drain together with a wire. We understand this is not an accurate depiction of the actual circuit, and have changed the main text to address this.

2. The DC equivalent of the circuit, used for tunneling current measurement (shown in Fig.1c), is analogous to gating the ferroelectric MoS_2 with respect to top graphene.

How would the electrostatic doping affect the inbuilt dipole moment in R-MoS_2 ? Does it depolarize the ferroelectric crystal?

Due to the band mismatch between the conduction band of MoS_2 and the graphite Fermi level of 0.3 eV, when a small bias is applied ($V_{sb} < 0.8$ V for the device in Fig. 2) the MoS_2 remains insulating. We therefore treat it like an additional tunnelling barrier (skewed by the field) in our analysis of both

experimental results and modelling. For the larger V_{sb} values the MoS₂ indeed becomes doped, and the direct tunnelling into the bilayer becomes significant. Our earlier studies show that such filling of the conduction band indeed leads to screening of the ferroelectric dipole, through it persists to finite densities (Weston et al. Nat. Nanotechnol).

3. The theory describing seeding size vs. favorable energy, based on competitive domain-wall stiffness, deformation energy, and electrostatic energy, lacks an introduction. No physical description has been given besides citing Ref.29. I believe an intuitive description and reasoning are very much needed to make the current manuscript self-contained and would help increase the comprehensibility among a broader class of readers.

We agree with the referee that the manuscript will benefit from a more general discussion of the modelling. This have now been included in the main text (see highlighted changes).

4. Domains of type 3-4 (in ref. to Fig.1) do not show any prominent hysteresis in di/dv (Fig. 3b-c). However, it is contrasting (at least partially) to earlier results. See, for example - Fig.1 Weston et al. Nat. Nanotechnol. 2022 -analogous to type 4; SI-Fig.S2 Deb et al. Nature 2022 -analogous to type 3. The question occurs: Do the di/dv plots indicate (i) NO sliding & NO hysteresis or (ii) only NO hysteresis or (iii) something else?

We fully agree with the referee that in both references, two partial dislocations are shown to merge and split again with the cycling of the electric field. One explanation we have is that when two partial dislocations collide, they do not necessarily zip into a perfect dislocation along the entire length of the boundary. Indeed, this can be seen in Weston et al. Fig. 1e,f where the resolution is sufficient to spot several pockets where the two partial dislocation have not merged and are tens on nms apart. Upon the field reversal, these pockets serve as seeds for the unzipping, hence the observed behaviour. This is in contrast to our current work, where an already formed perfect dislocation was selected for the device fabrication. Most likely we deal with a different case where there is no such “weak spots” along the domain wall to provide the seed. To acknowledge the referee’s comment we have added a sentence to the main text.

5. Please consider adding a summarized conclusion at the end.

We agree with the referee that the manuscript reads better with a conclusion – it has now been added.

6. Usually, such device fabrication involves various thermal annealing steps, both after tear-and-stack (before lithography) and after lithography. If this is the case, comments in supplementary would be helpful for the work to be followed and reproduced in the future.

We apologize for the omission and agree that such details are important. The annealing steps have now been described in the SI section 1.

7. Please have a look at Liang et al. Shear Strain-Induced Two-Dimensional Slip Avalanches in Rhombohedral MoS₂, Nano Lett. 2023, 23, 15, 7228. Consider citing it if you find it relevant and useful.

Thank you, the reference has now been included in the appropriate part of the introduction.

REVIEWER COMMENTS

Reviewer #1 (Remarks to the Author):

the authors have answered my questions and it could be published after citing 2 related papers as following:

J. Am. Chem. Soc. 144, 3949–3956 (2022)

Phys. Rev. B 108, L161406 (2023)

Reviewer #2 (Remarks to the Author):

With new experimental results and theoretical modeling, the authors have addressed most of my concerns. Nevertheless, I strongly suggest clarifying the feasible ways to enhance the ratio of tunneling conductance between two polarized states, at least in the discussion part. This is important to attract attentions of a wide readership.

Reviewer #3 (Remarks to the Author):

I appreciate the authors' efforts in addressing all the comments. They have satisfactorily addressed my comments/questions and incorporated the necessary changes. Consequently, I am pleased to recommend the manuscript for publication. Nevertheless, I have two subtle quires/recommendations.

1. Regarding the answer to Q2: The persistence of ferroelectric dipoles against state filling (up to $1E13 \text{ cm}^{-2}$, [Ref 18,21], rather than the voltage [0.8V, Main text Line 126-127]) determines the fundamental boundary for the application of such devices. Given the manuscript's title, this is a valuable information to add to the text.

2. I would appreciate remarks on the energetic and topological stability of partial and perfect dislocations in such samples.

REVIEWER COMMENTS

To all: We would like to thank the Referees once again for their helpful comments and constructive criticism during this process. We appreciate the time it has taken to review our manuscript and believe that the manuscript has been considerably improved as the result of their work. To address the last round of the comments, clarifications have been made to the main text and an additional SI Section 6 along with Figure S9 have been added. A point-by-point response to the reviewers' concerns follows.

Reviewer #1 (Remarks to the Author):

the authors have answered my questions and it could be published after citing 2 related papers as following:

J. Am. Chem. Soc. 144, 3949–3956 (2022)

Phys. Rev. B 108, L161406 (2023)

We thank the reviewer for their work, the papers are now cited in the main text.

Reviewer #2 (Remarks to the Author):

With new experimental results and theoretical modeling, the authors have addressed most of my concerns. Nevertheless, I strongly suggest clarifying the feasible ways to enhance the ratio of tunneling conductance between two polarized states, at least in the discussion part. This is important to attract attentions of a wide readership.

The relevant discussion has now been included in the main text (highlighted blue).

Reviewer #3 (Remarks to the Author):

I appreciate the authors' efforts in addressing all the comments. They have satisfactorily addressed my comments/questions and incorporated the necessary changes. Consequently, I am pleased to recommend the manuscript for publication. Nevertheless, I have two subtle quires/recommendations.

1. Regarding the answer to Q2: The persistence of ferroelectric dipoles against state filling (up to $1E13 \text{ cm}^{-2}$, [Ref 18,21], rather than the voltage [0.8V, Main text Line 126-127]) determines the fundamental boundary for the application of such devices. Given the manuscript's title, this is a valuable information to add to the text.

2. I would appreciate remarks on the energetic and topological stability of partial and perfect dislocations in such samples.

We thank the reviewer for valuable comments, both clarifications have now been included in the main text (highlighted in green)

REVIEWERS' COMMENTS

Reviewer #2 (Remarks to the Author):

The authors have addressed my concerns, thus I recommend its publication in Nature Communications.